# Diagnostic and Prognostic Significance of MiR-150 in Colorectal Cancer: A Systematic Review and Meta-Analysis

**DOI:** 10.3390/jpm10030099

**Published:** 2020-08-24

**Authors:** Daniel Sur, Claudia Burz, Shanthi Sabarimurugan, Alexandru Irimie

**Affiliations:** 111th Department of Medical Oncology, University of Medicine and Pharmacy “Iuliu Hatieganu”, 400015 Cluj-Napoca, Romania; 2Department of Medical Oncology, The Oncology Institute “Prof. Dr. Ion Chiricuta”, 400015 Cluj-Napoca, Romania; cburz@yahoo.fr; 3Department of Immunology and Allergology, University of Medicine and Pharmacy “Iuliu Hatieganu”, 400162 Cluj-Napoca, Romania; 4School of Biomedical Sciences, Faculty of Health and Medical Sciences, The University of Western Australia, Nedlands, WA 6009, Australia; shanthi.sabarimurugan@uwa.edu.au; 511th Department of Oncological Surgery and Gynecological Oncology, “Iuliu Hatieganu” University of Medicine and Pharmacy, 400015 Cluj-Napoca, Romania; airimie@umfcluj.ro; 6Department of Surgery, The Oncology Institute “Prof. Dr. Ion Chiricuta”, 400015 Cluj-Napoca, Romania

**Keywords:** miR-150, colorectal cancer, metastases, biomarker, diagnostic, prognostic

## Abstract

Although treatment options have improved, the survival and quality of life of colorectal cancer (CRC) patients remain dismal. Therefore, significant biomarker prediction may help to improve colorectal cancer patient’s prognosis profile. MiRNAs have come as an option because of their essential role in cancer initiation and progression by regulating several molecular processes. MiR-150 has different roles in cancer, but its function in CRC is still ambiguous. We undertook a systematic review and meta-analysis according to the Preferred Reporting Items for Systematic Reviews and Meta-Analyses (PRISMA) research criteria by interrogating several databases in order to assess the diagnostic accuracy and prognostic value of miR-150. Additionally, clinicalgov.org was scanned for possible trials. The literature was screened from inception to February 2020. A total of 12 out of 70 full-text articles were included in the meta-analysis. Among these, nine studies were included for diagnostic accuracy, and the remaining three were considered for prognostic significance of miR-150. With our results, miR-150 is an appropriate diagnostic biomarker, especially in serum and plasma, while the prognostic value of miR-150 was not statistically significant. The present study findings suggest that miR-150 has high specificity and sensitivity values as a potential diagnostic biomarker in colorectal cancer patients.

## 1. Introduction

Colorectal cancer (CRC) is a major health threat worldwide. In 2018, approximately 6.1% of all cancer cases diagnosed were colorectal cancers. Additionally, CRC is responsible for about 5.8% of cancer-related deaths [1]. This situates CRC as the third most diagnosed malignancy and the second leading cause of mortality related to oncological malignancies globally [2]. At initial evaluation, about 36% of the patients with CRC are diagnosed with lymphatic invasion and 20% with metastatic disease [3]. For early-stage disease, the five-year survival rate for CRC patients is 90%. On the other hand, in stage IV disease, the percentage drops to merely 10% [4]. This highlights the urgency of finding proper biomarkers for early diagnosis.

The cornerstones of colorectal cancer treatment remain surgical approach and chemotherapy. Recent therapeutic modalities have improved CRC prognosis. Part of this improvement in metastatic colorectal cancer (mCRC) is due to the addition of cytotoxic agents to targeted therapies such as anti-EGFR monoclonal antibodies (Cetuximab, Panitumumab) or antiangiogenic drugs targeting VEGF signaling (Bevacizumab, Ramucirumab). First and second-line therapies for metastatic colorectal cancers consists of the association of classical or modern chemotherapy regimens such as Oxaliplatin, Irinotecan, Capecitabine, or Trifluridine/Tipiracil to biologically targeted agents [5]. Regorafenib, a multitargeted tyrosine kinase inhibitor, comes as an option for patients with refractory metastatic colorectal cancer [6].

Currently, the clinical guidelines [7] do not suggest any predictive biomarkers for treatment response to drugs targeting VEGF-signaling. RAS wild type and BRAF status are the only consecrated biomarkers for choosing anti-EGFR therapy [8]. As recent data suggest that miRNAs might have crucial roles in the development and progression of CRC, we hypothesize these small molecules would act as specific targets for diagnosis, prognosis, and treatment [9,10]. Additionally, the tumor microenvironment represents another culprit for the challenge of finding specific drug targets [11]. A handful of trials proposed targeted therapies or immunotherapy for the treatment of metastatic CRC [12]. The responses to treatment vary significantly between colorectal cancer patients, and the need for standardizing treatments seems mandatory before the adequate standard care therapy has exhausted [13].

At a molecular level, CRC is considered to have a very complex mechanism, where most of the studies concentrate on coding or non-coding genes. MicroRNAs (miRNAs) are non-coding small RNA molecules that act as essential posttranscriptional gene expression regulators [14]. Many miRNAs, as well as miR-150, could act as an oncogene by promoting tumor initiation and invasion or as a tumor suppressor by inhibiting tumor growth and metastasis [15].

With accumulating evidence, miR-150 appears up-regulated in various human cancers such as hepatocellular cancer [16], cervical cancer [17], NSCLC [18], and colorectal cancer [19]. On the other hand, more recently, it has been suggested that down-regulated expression of miR-150 in conjunction with elevated Gli1 (glioma-associated oncogene homolog 1) is responsible for the epithelial-mesenchymal transition (EMT) that promotes invasion and metastasis in colorectal cancer cells [20]. Exosomal miR-150-5p acts as a potential non-invasive diagnostic and risk assessment tool for patients with colorectal cancer [21]. MiR-150-5p is associated with TP53 suppression in colorectal cancer and prevails prognostic attributes. This association appears to be involved in the invasion and migration processes and the suppression of apoptosis [22]. Based on a literature search by our team that was conducted in Pubmed to find functionally relevant miRNAs involved in modulating the invasive phenotype of CRC, we selected miR-150 as being a potential biomarker that is also exerting an influence on the tumor microenvironment. The search words were: miRNA, miR, microRNA, colorectal cancer, and invasive phenotype.

The objective of this systematic review and meta-analysis is to summarize the current knowledge regarding miR-150 and to evaluate its diagnostic and prognostic impact in patients with colorectal cancer.

## 2. Materials and Methods

### 2.1. Data Sources and Search Strategy

We undertook a systematic review and meta-analysis by adopting the Preferred Reporting Items for Systematic Reviews and Meta-Analyses (PRISMA) guidelines [23]. All the search steps were done according to the Cochrane handbook for systematic reviews [24]. To identify relevant studies, we investigated several online bibliographic databases (Cochrane Library, Embase, Medline, PubMed and Google Scholar) from the time of inception until February 2020. A further search was done in the clinicalgov.org database to identify any missing publications and to evaluate all current trials concerning miR-150 in CRC. The key-words for the search strategy were as follows using different combinations of the terms related to miR-150 and colorectal cancer diagnosis and prognosis: “miR 150,” “microRNA 150,” “miRNA 150,” “hsa-miR-150,” “colon cancer,” “colorectal cancer,” “rectal cancer,” “prognostic,” “diagnostic.” This strategy was applied to all retrieved studies. The last search was performed on 29 February 2020. Moreover, the citations of the retrieved articles were screened for possible relevant articles using a forward/backward reference search method. The retrieved publications were screened against the selection criteria.

### 2.2. Study Selection

Two authors (D.S. and C.B.) proceeded to read all the titles and abstracts of the retrieved publications for inclusion. If discrepancies appeared between the first two authors, the third author (S.S.) reviewed the articles and resolved any conflicts.

### 2.3. Inclusion and Exclusion Criteria

The inclusion criteria in this review was as follows: (1) Randomized control trials, case-control, cross-sectional, and cohort studies; (2) Only peer-reviewed human studies published in English with available full text were considered; (3) Studies with 18-year-old patients or above diagnosed with colorectal or rectal cancer; (4) The studies that investigated at least one diagnostic and prognostic measure and mentioned the up or down-regulated expression levels of miR-150 in colorectal cancer from tissue/serum/plasma samples; (5) For the meta-analysis, we selected articles that had measured hazard ratios (HR) and 95% confidence intervals (CI) for the prognostic evaluation of miR-150; (6) For the diagnostic value of this review, we extracted and analyzed AUC (Area Under the Curve) measurements.

The exclusion criteria for the articles were as follows: (1) non-English publications, conference/meetings abstracts; (2) Articles published on animals/cell lines, non-full text, reviews and meta-analysis papers, case reports, letters to the editor; (3) Articles that failed to study the miR-150 expression in human samples; (4) Articles using fewer than 10 samples.

### 2.4. Data Extraction

The authors (D.S., C.B., and S.S.) extracted data from the included papers and entered them in an Excel worksheet. Any disagreement was resolved through discussion between the authors. The following information were recorded in the Excel form: data about the publication (year of publication, country, first author’s name), population (size of the population, baseline characteristics, sample type, tumor data (anatomic location), method and sample of miR-150 measurement (up-regulated/down-regulated), and statistical analysis measures (HRs associated with 95% CI for OS). Additionally, diagnostic and prognostic measures were extracted. The diagnostic potential of miR-150 considering sensitivity and specificity was marked using the ROC (Receiver Operating Characteristics) curve method. The prognostic potential of the biomarker was reported by evaluating the Kaplan-Meier curves.

### 2.5. Quality Assessment

For assessing the quality of the included studies, we used the Quality Assessment of Diagnostic Accuracy Studies (QUADAS) II tool [25]. The tool consists of 14 questions, and the answer could be appraised as Yes, No, or Unclear, matching to a score of 1, 0, or 0, respectively. The issues addressed by this tool include: the selection criteria of the study groups in each study, the comparability of groups, and the establishment of exposure/outcome of interest for case-control/cohort studies. This tool offers a succinct and straightforward, yet comprehensive analysis of the quality of each included study, and allows us to visualize it with clarity.

### 2.6. Meta-Analysis Assessment

Analyses were conducted using Review Manager Software 5.3.5 and Comprehensive Meta-Analysis Software version 3.0. The meta-analysis was performed using the comprehensive meta-analysis (CMA) software for the extracted hazard ratio (HR) values and 95% confidence intervals (CIs) of the articles and Kaplan Meier curves. Additionally, random effects models were used for the meta-analysis. Statistical heterogeneity between studies was evaluated by performing the Higgin’s I^2^ statistic and the Tau^2^ value [26]. A *p*-value less than 0.05 was considered as reflecting heterogeneity, and the random effects model was applied as well. For summarizing the HR estimates, we used the forest plot drawings.

## 3. Results

### 3.1. Search Results

Seventy articles were identified through an online database search. Thirty articles were removed as duplicates and 20 were excluded after screening. The remaining 20 articles were reviewed in detail and eight were excluded as they did not provide enough diagnostic and prognostic results to be included. Finally, 12 articles [21,27,28,29,30,31,32,33,34,35,36,37] were selected and included in the qualitative and quantitative analysis. From these 12 articles, one [37] article was excluded due to a lack of numerical data and 11 articles [21,27,28,29,30,31,32,33,34,35,36] were eligible for the meta-analysis. In Figure 1, the study selection and article screening through appropriate flow diagram is demonstrated.

### 3.2. Characteristics of the Selected Studies

The basic characteristics from the 12 articles [21,27,28,29,30,31,32,33,34,35,36,37] are listed in Table 1. Nine studies [21,27,28,29,30,31,32,33,37] were diagnostic studies, while three studies [34,35,36] were prognostic studies. One diagnostic [37] study met the inclusion criteria but was finally excluded due to lack of numerical data report. Therefore, we included it in the qualitative synthesis, but not in the meta-analysis.

Five of the studies were conducted in China [21,30,32,35,36], two in Spain [28,29], and one of each in Czech Republic [31], Turkey [37], Slovakia [33], Italy [34], and Japan [27]. In total, these studies included 1050 cancer patients and 734 controls. The staging of the patients included in this review ranged from stage 0 (Tis) to stage IV. The type of biological samples collected were serum (n = 5), plasma (n = 3), tissue (n = 3), and peritoneal lavage (n = 1) among the selected studies. All of the studies used Q-RT-PCR for measuring the expression of miRNA.

### 3.3. MiR-150 as A Diagnostic Biomarker

Six studies [21,28,29,30,32,33] also reported sensitivity and specificity for the diagnostic value of miRNA150. The minimum reported sensitivity and specificity were 57.75 and 56.25%, respectively, whereas the maximum reported sensitivity and specificity were 93.6 and 90%, respectively.

The performance of miRNA-150 as a diagnostic biomarker was assessed using AUC. Our meta-analysis included eight relevant studies [21,27,28,29,30,31,32,33]. The estimated AUC ranged from 0.632 to 0.978 (Figure 2). Five studies [21,27,30,31,32] investigated miRNA in serum, two [28,33] in plasma, and one [29] in peritoneal lavage. The *p*-values according to the analysis were less than 0.001, meaning that they demonstrated a significant diagnostic value of miR-150. Therefore, we used the random-effects model and found an AUC of 0.830, 95% CI [0.761–0.898] (Figure 3). Pooled analysis was heterogeneous (I^2^ = 89%).

Serin Akbayir et al. [37] assessed the early diagnosis potential of up or down-regulated miRNAs expression of colorectal cancer patients compared to the control group. We excluded this article from the meta-analysis because the study did not provide sufficient numerical data. This study remained only in the qualitative analysis. The study results reported that the miR-150-5p, miR-30a-5p, miR-34a-5p, and miR-195-5p could be beneficial in the early diagnosis of colorectal cancer patients.

### 3.4. MiR-150 as A Prognostic Biomarker

Our meta-analysis of prognostic studies included three studies [34,35,36]. The HR value of OS from all three studies ranged from 0.48 to 1.92. The random-effects model reported a pooled HR of 1.05, 95% CI [0.57–1.94]. We applied the random effect model to evaluate the heterogeneity of the studies based on the Higgins I^2^ test which is represented as 25%, 50%, and 75% percentage corresponding to a low, moderate, and high level of heterogeneity, respectively. Our studies resulted in an I^2^ test of 83%, which corresponds with a high level of heterogeneity.

In the study by Yanlei Ma et al. [35], decreased expression of miR-150 was linked with shorter survival and a weaker response to adjuvant chemotherapy in patients with CRC.

In the study by Pizzini et al. [34], miR-150 is significantly down-regulated in primary tumoral and metastatic tissue compared with normal mucosa of the colon. The third study from our meta-analysis [36] showed that miR-150-5p is down-regulated in CRC tissue and is negatively related with TNM staging, pathological lymph node status, and overall survival. Additionally, VEGFA was found to be a direct target gene of miR-150-5p, leading to new possibilities in treatment development.

The forest plot displayed an HR of 1.05, 95%CI [0.57–1.94], with a *p*-value of 0.86 (Figure 4), which signifies a low predictive value for CRC patient mortality. The test for overall effect displayed a Z-value of 0.17. Overall, in our meta-analysis, the down-regulated expression of miR-150 did not manifest a prognostic role of survival outcomes for CRC patients.

### 3.5. Bias Assessment and Applicability Judgments

For quality assessment, we individually assessed the risk of bias and applicability judgments using QUADAS-II [25]. The following listed items were assessed as being high, unclear or low concerning bias: (1) patient selection; (2) index test; (3) reference standard; (4) flow and timing. Two reviewers evaluated the quality of selected studies and our assessment revealed that most of the included studies received a low or unclear risk of bias. The risk of bias graph and summary are shown in Figure 5.

## 4. Discussion

MiRNAs play a crucial role in the regulation of one-third of all human genes and are closely associated with the carcinogenesis process [38]. Some published studies refer to the role of miRNA-150 in various cancers [39,40], but the significance of prognosis of survival and the diagnostic accuracy of miR-150 has not been covered. This systematic review and meta-analysis synthesized evidence for the diagnostic and prognostic value of miRNA150 for colorectal cancer by involving 12 studies [21,27,28,29,30,31,32,33,34,35,36,37], in which 1050 CRC patients were used to determine the diagnostic and prognostic efficacy of miRNA150. Overall, it was found that it is a promising diagnostic marker, especially in serum and plasma of colorectal cancer patients.

Generally, our findings reveal that miRNA150 is not a statistically significant biomarker for the prognosis of colorectal cancer patients. This may be due to the low number of studies included in the meta-analysis to evaluate the prognostic significance.

As mentioned before, low level expression of miR-150 in CRC patients has been reported in other articles as a probable diagnostic and predictive biomarker [41]. Chen Li et al. [42] highlighted that the decreased expression of miR-150 in clinical CRC samples could be a marker of poor prognosis for this category of patients. Our study completes in a systematic manner the already existent evidences about miR-150 biomarker potential updating the data and encompassing several databases for accurate analysis.

A total of eight diagnostic studies [21,27,28,29,30,31,32,33], with a total of 616 patients with colorectal carcinoma compared with 496 cases as a control group, showed that miRNA150 is statistically significant in the group with colorectal cancer patients (*p*-value < 0.001, 95% CI [0.761–0.898]). Furthermore, the AUC value is a high indicator of a diagnostic test, and analysis showed the AUC to be 0.830, which also indicates that miRNA150 could be a valid biomarker for colorectal cancer detection. Though the variance and degrees of freedom are high, the supporting evidence of *p*-value and I-squared heterogeneity above 75% brings the clinical value of miR-150. Additionally, the noteworthy ROC value should be considered, which is the main parameter to understand the diagnostic accuracy of any biomarker. Hence, we consider miR150 has appreciable diagnostic value to boost up the clinical significance in CRC patients. From an analysis of a set of 42 differentially expressed miRNAs in CRC patients and healthy subjects, miR-150 appeared to be down-regulated in tissue samples and was able to differentiate cancer tissue from adjacent normal mucosa [43].

Regarding prognosis, three studies [34,35,36] with a total of 397 patients with CRC were analyzed. It was found that miRNA150 is not a statistically significant predictor of poor health condition (HR 1.05, 95% CI [0.57–1.94]). Most of the studies reported that this miRNA has a low expression in cancer patients than in the controls. This finding is in line with other studies analyzing the role of other miRNAs in colorectal cancer [44,45]. In a recent meta-analysis, low expression of miR-143 and miR-145 is correlated with a poor survival prognosis [46]. Additionally, miR-21 appears to be a valuable prognostic marker for colorectal, pancreatic and esophageal cancer. This miRNA is correlated with a poor overall survival and also a poorer disease-free survival [47].

Patients with colorectal cancer are highly prone to relapse or mortality due to delayed diagnosis [48]. Although serious progress has been made concerning treatment options, non-invasive cost-efficient specific biomarkers are needed for the early detection of CRC patients [49]. MiRNAs have critical cellular function, thus, the researchers are evaluating the relevant molecular pathways to find solutions for better estimation of cancer progression. Takaaki Masuda et al. reviewed the clinical significance of miRNAs as biomarkers in colorectal cancer patients. The studies analyzed consisted of small-scale trials and one meta-analysis and exhibited that miRNAs have biomarker potential and also can assist in new drug development due to their capacity to modify tumorigenesis and tumor development in CRC [50]. Shanthi Sabarimurugan et al. has studied the miRNAs roles as prognostic biomarkers for patients with stage II CRC. The results showed that miRNAs could bring prognostic value to stage II CRC patients, and that specific miRNAs can have a dual role as a biomarker and therapeutic target [51].

### Study Strengths and Limitations

The strengths of this study consist of the following: the search strategy for this review is comprehensive due to data collection from multiple databases, 12 unique studies were included, the analysis was done using more than one method, and the quality assessment was done by QUADAS II tool [25]. All steps of the research protocol were done corresponding to the Cochrane Handbook for systematic reviews [24] and a PRISMA checklist [23].

As far as we know, this is a one of its kind systematic review and meta-analysis of miR-150 diagnostic and prognostic potential for colorectal cancer. Furthermore, identifying miRNAs that can be used as an early non-invasive diagnostic tool and also as an indicator of poor prognosis in CRC patients can open new doors for patient selection and individualized treatments.

The limitations of the present review include restriction to exclusively English studies. Additionally, the small sample sizes of the retrieved studies elevated the risk of publication bias. Moreover, Shui-Lan Zou’s [30] study was excluded from prognostic analysis due to reporting odd ratio for the prognostic data. We also could not resolve the heterogeneity detected in the analysis of the prognostic studies as the number of included studies was small.

We recommend conducting further studies with clear protocols and large sample sizes to detect the evidence for prognosis and to add extra clear evidence for diagnosis. Additionally, performing other systematic reviews and meta-analysis without language restriction will bring a new perspective to this field.

## 5. Conclusions

In conclusion, in the present systematic review and meta-analysis, we studied the diagnostic and prognostic role of miR-150 in colorectal cancer patients. Our study revealed that miR-150 could be beneficial as a diagnostic biomarker for colorectal cancer patients, while no significant evidence was found in miR-150 for prognosis. However, future validation is mandatory before we confirm miR-150’s biomarker applicability. Even though miRNAs significantly impacted the CRC patient’s survival outcomes, the small number of studies involved in the analysis have reduced their role in day-to-day clinical use. We propose that further clinical large-scale trials and longitudinal studies concerning miR-150 should be conducted.

## Figures and Tables

**Figure 1 jpm-10-00099-f001:**
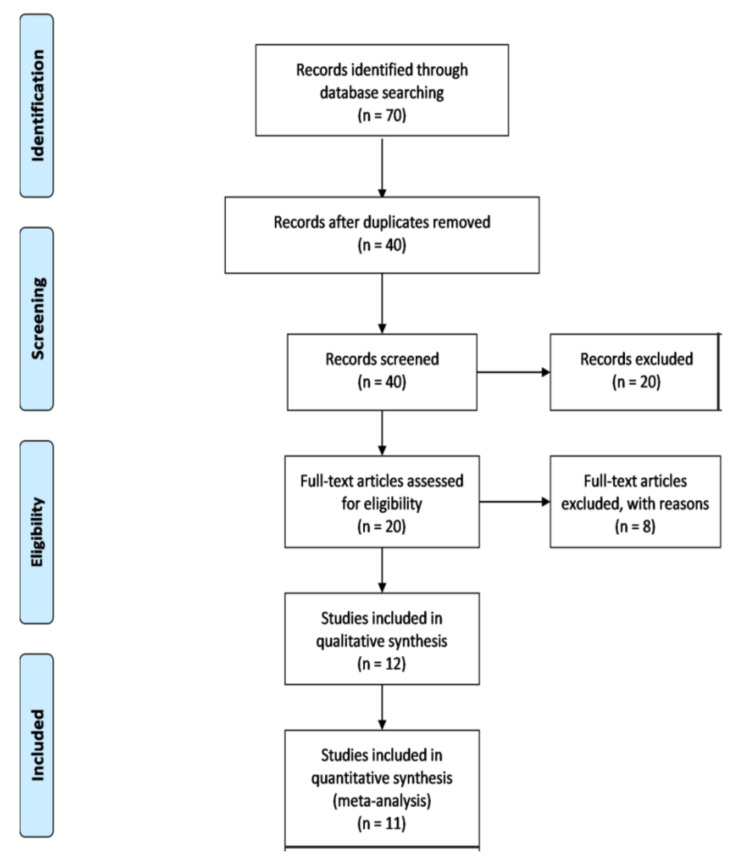
PRISMA flow diagram of the study selection process.

**Figure 2 jpm-10-00099-f002:**
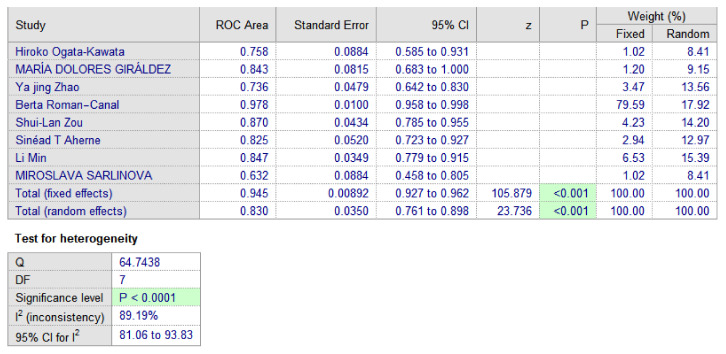
Summary of meta-analysis for diagnostic studies.

**Figure 3 jpm-10-00099-f003:**
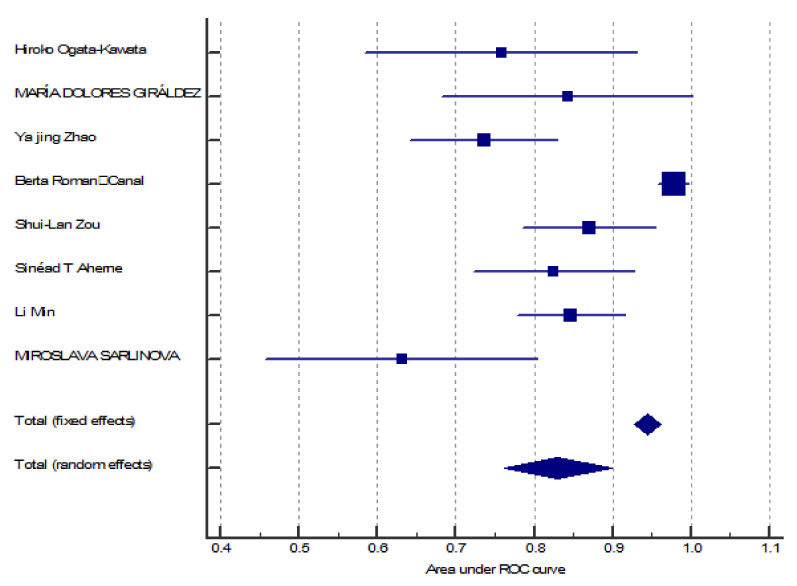
Forest plot for meta-analysis of AUC using a random-effects model.

**Figure 4 jpm-10-00099-f004:**
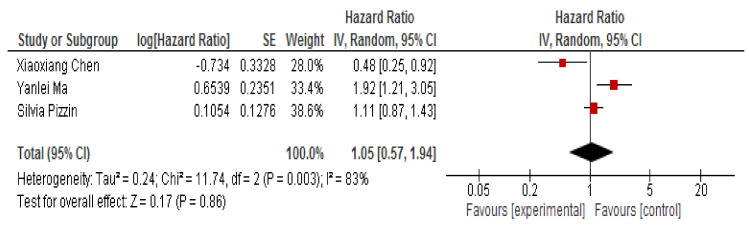
Forest plot for meta-analysis of HR using a random-effects model.

**Figure 5 jpm-10-00099-f005:**
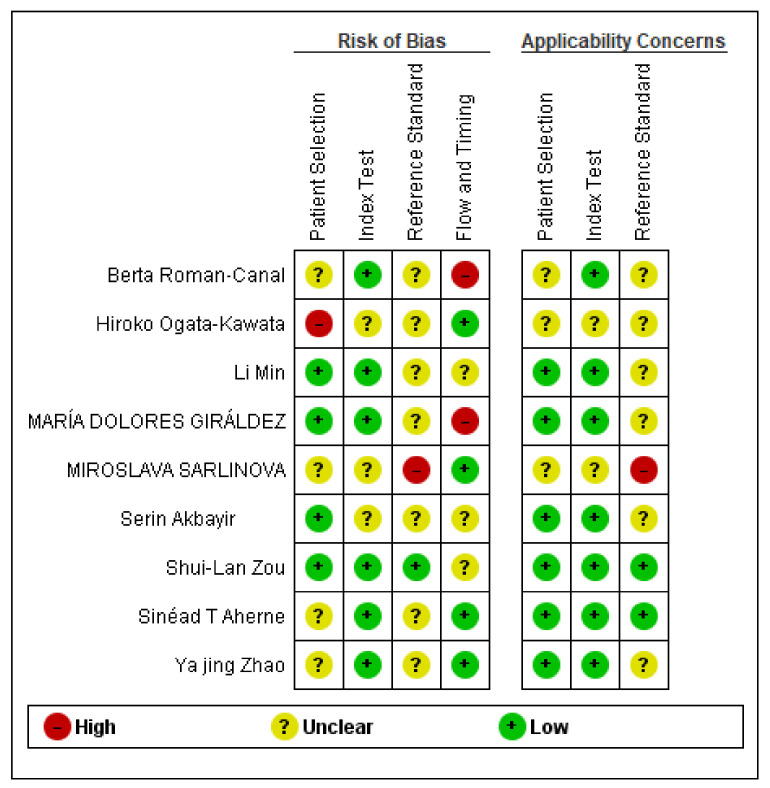
Risk of Bias and Applicability Judgments according to QUADAS-II.

**Table 1 jpm-10-00099-t001:** Basic characteristics of the articles included.

ID	Year	Country	No. of Patients	No. of Controls	MiRNA	Sample	Method for Detection	Reported Value	Expression Status
Hiroko Ogata-Kawata [27]	2014	Japan	88	11	150	Serum	RT-PCR	AUC, 95% CI	Up-regulated
MARÍA DOLORES GIRÁLDEZ [28]	2013	Spain	63	73	150	Plasma	RT-PCR	AUC, 95% CI	Up-regulated
Ya Jing Zhao [21]	2019	China	169	155	150-5p	Serum	RT-PCR	AUC, 95% CI	Down-regulated
Berta Roman-Canal [29]	2019	Spain	25	25	150-5p	Peritoneal lavages	RT-PCR	AUC, 95% CI	Up-regulated
Shui-Lan Zou [30]	2019	China	133	60	150-5p	Serum	QRT-PCR	AUC, 95% CI	Down-regulated
Sinéad T Aherne [31]	2015	Czech Republic	52	82	150	Serum	QRT-PCR	AUC, 95% CI	Down-regulated
Li Min [32]	2019	China	15	10	150-3p	Serum	QRT-PCR	AUC, 95% CI	Down- and up-regulated
MIROSLAVA SARLINOVA [33]	2016	Slovakia	71	80	150	Plasma	QRT-PCR	AUC, 95% CI	Down-regulated
Serin Akbayir [37]	2013	Turkey	37	238	150-5p	Plasma	QRT-PCR	-	Down-regulated
Silvia Pizzin [34]	2013	Italy	46	-	150	Tissue	QRT-PCR	HR, 95% CI	Down-regulated
Yanlei Ma [35]	2012	China	239	-	150	Tissue	QRT-PCR	HR, 95% CI	Down-regulated
Xiaoxiang Chen [36]	2018	China	112	-	150-5p	Tissue	QRT-PCR	HR, 95% CI	Down-regulated

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
