# Peer review of "Diagnostic and Prognostic Significance of MiR-150 in Colorectal Cancer: A Systematic Review and Meta-Analysis"

_jpm, 2020, doi:10.3390/jpm10030099_

Round 1

Reviewer 1 Report

The author is making a bold statement about the diagnostic value of a liquid biopsy without addressing several issues such as cancer staging. The author briefly mentions that stage at diagnosis correlates with prognosis but does not address whether the articles reviewed include data regarding staging. I think it would greatly strengthen the paper to address staging in the article since it is so critical for determining patient outcome. Even if it is just to say that the lack of staging data is a limitation of the study I feel that it is important to address. 

Overall I think the paper was well-written and the methods for article selection were clearly outlined. There were a few instances where the wording was a little awkward which made it difficult to read.

I found a few typos and awkwardly-worded sentences throughout the paper as included below:

Line 29 - typo, "form" should be "from"

Lines 48-51 - the wording is confusing

Line 72 - consider changing "including" to "and"

Lines 77-78 - consider clarification of wording

Line 147 - "this" should by "these"

Line 190-191 - Is this sentence its own paragraph or should it be included with the paragraph starting on line 192 (page-break makes this unclear)

Line 213-214 - consider revision of sentence

Line 214-216 - consider revision of sentence

Line 271-273 - wording is confusing, consider revision

Author Response

We thank you for your pertinent observations regarding our review, and we consider that addressing your request we improve the quality of our manuscript entitled:” Diagnostic and Prognostic Significance of MiR-150 in Colorectal Cancer: A Systematic Review and Meta-Analysis.”

Reviewer 2 Report

Sur et al. performed a careful review and meta-analysis of diagnostic accuracy (9 articles) and prognostic significance (3 articles) of miR-150, selected from 70 full-text articles. The authors conclude that miR-150 is an appropriate diagnostic biomarker, especially in serum and plasma, but evidence for prognostic value is lacking. Overall, this review is a nice contribution, reviewing this particular topic. 

Some issues may need to be addressed.

  1. The justification for selecting only miR-150 can be strengthened (in view of a very recent review of multiple miRs).
  2. Variance is quite large between studies: 3 studies report miR150 to be upregulated, and 9 indicate downregulation. It appears to differ as a function of where miR150 was measured; sensitivity and specificity vary over a wide range.
  3. ROC area estimate appears to be largely driven by the results of the Berta Roman-Canal study (0.978). It would be appropriate to discuss why this particular study gave such high estimates, and why this result can be accepted in driving the ROC estimate.
  4. The authors state that their analysis demonstrates “significant diagnostic value of miR-150”. Yes, statistically this is correct but given the variance, what is the clinical value? Some discussion is needed.
  5. There are a few awkward sentences throughout that should be edited., for example “MiR-150-5p is associated with suppression of TP53 in colorectal cancer and prevails prognostic attributes.” And “Besides, the down-regulated expression of performed  miRNAs in CRC might be an eye opener for therapeutic development for this disease.” The latter is an offhand remark that requires more discussion or should be deleted.

Author Response

(The authors gave the same response as above.)
